# Hierarchical Action of Mulberry miR156 in the Vegetative Phase Transition

**DOI:** 10.3390/ijms22115550

**Published:** 2021-05-24

**Authors:** Hongshun Li, Yiwei Luo, Bi Ma, Jianqiong Hu, Zhiyuan Lv, Wuqi Wei, Haiye Hao, Jianglian Yuan, Ningjia He

**Affiliations:** State Key Laboratory of Silkworm Genome Biology, Southwest University, Beibei, Chongqing 400715, China; hong_shun_li@163.com (H.L.); luoyiwei12@swu.edu.cn (Y.L.); mbzls@swu.edu.cn (B.M.); jianghuwuji@163.com (J.H.); lvzy@email.swu.edu.cn (Z.L.); swuwuqi@email.swu.edu.cn (W.W.); 18375722578@163.com (H.H.); yuanjiangl@swu.edu.cn (J.Y.)

**Keywords:** age pathway, mir156s, mulberry, perennial woody plants, vegetative phase transition

## Abstract

The vegetative phase transition is a prerequisite for flowering in angiosperm plants. Mulberry miR156 has been confirmed to be a crucial factor in the vegetative phase transition in *Arabidopsis thaliana*. The over-expression of miR156 in transgenic *Populus × canadensis* dramatically prolongs the juvenile phase. Here, we find that the expression of mno-miR156 decreases with age in all tissues in mulberry, which led us to study the hierarchical action of miR156 in mulberry. Utilizing degradome sequencing and dual-luciferase reporter assays, nine *MnSPL*s were shown to be directly regulated by miR156. The results of yeast one-hybrid and dual-luciferase reporter assays also revealed that six MnSPLs could recognize the promoter sequences of mno-miR172 and activate its expression. Our results demonstrate that mno-miR156 performs its role by repressing MnSPL/mno-miR172 pathway expression in mulberry. This work uncovered a miR156/SPLs/miR172 regulation pathway in the development of mulberry and fills a gap in our knowledge about the molecular mechanism of vegetative phase transition in perennial woody plants.

## 1. Introduction

During their post-embryonic development, plants need to go through three vital developmental processes from seed to mature plant, including seed germination, vegetative phase transition (juvenile-to-adult transition), and flower transition [1]. Vegetative phase transition determines the time of plant vegetative growth, and plants must go through sufficient vegetative growth from juvenile to maturity and eventual reproductive growth [2].

*SPL* genes have been shown to be repressed by miR156, and the miR156/SPLs/miR172 regulation pathway in vegetative phase transition was systematically revealed in *Arabidopsis thaliana* [3]. From seedlings to mature plants, the expression level of miR156 gradually decreased, while the expression levels of *SPLs* and miR172 gradually increased in model annual forbs, such as *A. thaliana* [3], *Nicotiana benthamiana* [4,5], and rice [6,7]. In *A. thaliana*, the roots were not involved in vegetative phase transition regulation; however, sugar in the leaves could trigger the vegetative phase transition by the suppression of miR156 expression [4,8,9]. Perennial woody plants had significantly longer vegetative growth times than annual plants, thus it is of more pragmatic value to study the vegetative phase transition mechanism of perennial woody plants [10]. Compared with annual plants, miR156 and miR172 showed similar expression trends in vegetative phase growth in some perennial woody plants such as *Acacia confusa*, *Acacia colei*, *Eucalyptus globulus*, *Hedera helix*, *Quercus acutissima*, *P. × canadensis* and some tropical/subtropical trees, and the over-expression of miR156 in *P. × canadensis* delayed its vegetative phase change [11,12]. Researchers also found that the expression of *SPLs* decreased under heavy fruit load in *Citrus clementina*, and Citrus *SPL* was able to promote flowering independent of photoperiod in *Arabidopsis* [13]. EjSPL4, EjSPL5, and EjSPL9 could significantly activate the promoters of *EjSOC1-1*, *EjLFY-1*, and *EjAP1-1* in loquat and the over-expression of *EjSPL3*, *EjSPL4*, *EjSPL5*, and *EjSPL9* in *A. thaliana* could promote obvious flowering [14]. However, the hierarchical action of miR156 in vegetative phase growth in perennial woody plants is still unknown.

Mulberry is a typical perennial woody plant with very important economic and medicinal values [15,16]. Mulberry trees have been utilized by humans for thousands of years, the leaves are used for raising silkworms, and the fruit is consumed fresh [17]. The medicinal value of mulberry has been explored, and abundant flavonoids and polysaccharides have been detected, especially in the leaves and fruits [18,19]. Mulberry plants also show strong resistance to abiotic stresses, and this crop can significantly improve the eco-environment [20]. Like other perennial woody plant breeding [21], a long juvenile period limits the mulberry resource development and utilization. In recent years, the mulberry genome sequencing has been gradually completed [16,22] and high-throughput sequencing of small RNA also has been finished in some varieties of mulberry [23,24]. All of these have laid a foundation for basic research on mulberry.

The molecular mechanism of the juvenile-to-adult transition in mulberry, however, is completely unknown. In this study, seven miR156s (mno-miR156a to mno-miR156g) and six miR172s (mno-miR172a to mno-miR172f) were identified using small RNAs high-throughput sequencing. Fourteen *MnSPL* genes were cloned. Our data revealed that miR156c directly repressed the transcription levels of nine *MnSPL* genes, including *MnSPL2*, *MnSPL15*, and *MnSPL16A*. Six *MnSPL* genes (*MnSPL2*, *MnSPL8*, *MnSPL10A*, *MnSPL10B*, *MnSPL15*, and *MnSPL16A*) could recognize the GTAC *cis*-element in the promoter region of mno-miR172a and activate the expression of mno-miR172a based on yeast one-hybrid experiments. An existing miR156/SPLs/miR172 pathway involved in the mulberry vegetative phase transition has been proposed.

## 2. Results

### 2.1. Mulberry miR156 Is Differentially Expressed in Juvenile and Mature Phase

In mulberry, the miR156 family has seven members named miR156a–g with mno-miR156c having a prominently higher expression than other mno-miR156s (Table 1). Sequence alignment and phylogenetic tree analysis showed that mulberry miR156s shared high sequence similarity, except miR156a. For example, there are only four base differences in miR156b–g (Figure 1a,b).

The Northern blot showed that miR156 had higher expression levels in young leaves from juvenile phase mulberry than in young leaves from mature phase mulberry (Figure 1c). Small RNA RT-qPCR revealed that the expression of miR156 gradually decreased with age in the roots, barks, and leaves of mulberry (Figure 1d). These results suggested that miR156 might be associated with vegetative phase transition in mulberry.

### 2.2. Identification of the miR156 Target Genes in Mulberry

Our results from in silico prediction showed that 115 genes from the mulberry genes database (https://morus.swu.edu.cn/, accessed on 3 March 2019) were selected as the presumptive miR156 target genes (Appendix A). GO analysis showed that over half of the presumptive miR156 target genes were clustered into the secondary level “binding” group in the first level molecular function group (Figure 2).

For further analysis of these presumptive target genes, we counted those genes whose expectation values were less than 2.5, and found seven *SPL* genes (*Morus010792*, *Morus010123*, *Morus015493*, *Morus018032*, *Morus014488*, *Morus017456*, and *Morus026457*), three *WD40 repeat-like* genes (*Morus026689*, *Morus002371*, and *Morus011698*), one *fructose 1,6-diphosphatase* gene (*Morus017876*) and two genes (*Morus021290* and *Morus006566*) with no annotation (Table 2). The results from the degradome sequencing verified seven *SPL* genes, including *MnSPL2*, *MnSPL4*, *MnSPL6*, *MnSPL13*, *MnSPL15*, *MnSPL16A,* and *MnSPL16B*, with the *p* values less than 0.05. The other genes with *p* values more than 0.05 were not considered as miR156 target genes (Figure 3 and Table 3).

Except for *MnSPLs*, no other genes were identified by degradome sequencing, which demonstrated that the *SPLs* genes were the main or exclusive target genes of miR156 in mulberry. Moreover, our results from the RT-qPCR showed that all of *MnSPLs* except for *MnSPL2* had a higher expression level in elder leaves, suggesting the opposite expression profile to mno-miR156 (Figure 4a,b).

Next, all of *MnSPL* genes predicted as miR156 target genes by psRNATarget were tested using dual-luciferase assays. Compared to the CK group, the pGreenII 0800 LUC vector with the supposed miR156 splice site of six *MnSPLs* (*MnSPL2*, *MnSPL7*, *MnSPL14*, *MnSPL15*, *MnSPL16A*, and *MnSPL16B*) showed significantly lower luciferase activity when miR156 was over-expressed (Figure 4c,d). We integrated the results of degradome sequencing and dual-luciferase assays, and identified nine *MnSPLs* (*MnSPL2*, *MnSPL4*, *MnSPL6*, *MnSPL7*, *MnSPL13*, *MnSPL14*, *MnSPL15*, *MnSPL16A*, and *MnSPL16B*) as miR156 target genes in mulberry (Figure 4e).

### 2.3. MnSPLs Bound to the Promoter of mno-miR172a and Promoted Its Expression in Mulberry

The result of the BLASTN and HMMER search showed there were 15 *MnSPL* genes in mulberry (Appendix A). All of the 15 *MnSPL* genes had the SBP conserved domains (Appendix A). The neighbor-joining (NJ) phylogenetic tree for the *SPL* genes from *M. notabilis*, *P. trichocarpa*, and *A. thaliana* showed that the *MnSPL* genes were separately distributed to five groups (Appendix A). Tissue expression profiles showed that *MnSPL* genes were differentially expressed in five tissues (winter bud, male flower, root, branch bark, and leaf). *MnSPL6*, *MnSPL8*, *MnSPL10*, *MnSPL15*, and *MnSPL16A* had significantly high expression in winter buds, while *MnSPL2*, *MnSPL7*, and *MnSPL16B* had much higher expression in male flowers. *MnSPL1*, *MnSPL13*, and *MnSPL14* were mainly expressed in the root, and *MnSPL4* and *MnSPL12* showed much higher expression in the leaf and branch bark, respectively (Appendix A). Six highly conserved mno-miR172s were identified in mulberry (Figure 5a,b), and the expression of mno-miR172a was significantly higher than the mno-miR172s in mulberry roots, barks, and leaves (Table 1).

The miR172 had a higher expression level in 9-month-old leaves than in younger leaves, which indicated that miR172 might be involved in the regulation of vegetative development in mulberry (Figure 5c). GUS activity was detected in CK, OE-156, and STTM156cd samples, indicating that genes were over-expressed by transient transgenic technology (Figure 6a,b). The over-expression of miR156 in mulberry leaves significantly decreased the expression of miR172 and *MnSPL* genes, except for *MnSPL1* and *MnSPL12*, while the down regulation of miR156 by STTM in mulberry seedlings significantly increased the expression of miR172 and most *MnSPL* genes except for *MnSPL1*, *MnSPL8*, *MnSPL12,* and *MnSPL14* (Figure 6).

We wondered if *MnSPLs* acted as upstream regulators of mno-miR172 and how *MnSPLs* functioned in this process. Thus, yeast one-hybrid assays were carried out to identify which *MnSPL* genes could recognize the predetermined GTAC elements in the mno-miR172a promoter. Except for three genes (*MnSPL6*, *MnSPL8*, and *MnSPL12*), which were toxic in yeast, the remaining 11 *MnSPL* genes were tested by the yeast one-hybrid assay, and six of those genes (*MnSPL2*, *MnSPL7*, *MnSPL10A*, *MnSPL10B*, *MnSPL15*, and *MnSPL16A*) could grow normally on the –Leu/A^150^ screening medium (Figure 7a,b).

Then, we used dual-luciferase assays to further verify these predicted miR156 target *MnSPL* genes (Appendix A) and found that six *MnSPL* genes (*MnSPL2*, *MnSPL8*, *MnSPL10A*, *MnSPL10B*, *MnSPL15*, and *MnSPL16A*) could increase luciferase activity. Taking the results of the yeast one-hybrid and dual-luciferase assays into consideration, we confirmed that six *MnSPL* genes (*MnSPL2*, *MnSPL8*, *MnSPL10A*, *MnSPL10B*, *MnSPL15*, and *MnSPL16A*) directly bound to the promoter of mno-miR172a and promoted its expression in mulberry.

## 3. Discussion

Woody plants were favorable systems for studying vegetative phase change because the stability and prolonged duration of the various stages of shoot development make these phases easy to observe and characterize [25]. However, the stability and prolonged duration of the vegetative stages limit breeding and genetic research in perennial woody plants. The miR156 over-expression can suppress vegetative phase transition and promote vegetative growth in poplar [11]. To date, however, researchers are still unclear about the molecular mechanism of vegetative phase development in perennial woody plants due to the limitations of the genetic research. Therefore, understanding the molecular mechanism of this process in perennial woody plants has very important research and practical value.

The sequence of miR156 was shown to be highly conserved among angiosperms [26]. Small RNA sequencing in mulberry showed that the expression level of mno-miR156c in three tissues (leaves, bark and male flowers) was significantly higher than other miR156 family members, and mno-miR156c also was shown to have the same sequences as miR156 found in *A. thaliana*, *G. max*, and *M. domestica* [23]. Similar to the discovery in *A. thaliana* [3] and *P. trichocarpa* [11], miR156 was expressed at higher levels in juvenile mulberry tissues than in mature mulberry tissues, which suggested that this conserved miRNA was probably involved in the vegetative phase transition in mulberry. In *A. thaliana*, *AtSPLs* were downstream factors of miR156-regulated vegetative stage development [3,27,28]. The *SPL* family gene function as promoters of vegetative phase transition and are distributed across angiosperms [29,30,31,32]. In total, there are 16 *AtSPL* genes in *A. thaliana* and 10 members (*AtSPL2*, *AtSPL3*, *AtSPL4*, *AtSPL5*, *AtSPL6*, *AtSPL9*, *AtSPL10*, *AtSPL11*, *AtSPL13*, and *AtSPL15*) were the miR156 target genes [3]. A total of 15 *MnSPL* genes were predicted by BLASTN and HUMMER modeling in MorusDB, and 14 gene sequences with an SBP-box domain were cloned successfully (*MnSPL10* has two alternative spliceosomes named *MnSPL10A* and *MnSPL10B*, but *MnSPL3* and *MnSPL5* were not cloned). A total of nine *MnSPLs* (*MnSPL2*, *MnSPL4*, *MnSPL6*, *MnSPL7*, *MnSPL13*, *MnSPL14*, *MnSPL15*, *MnSPL16A*, and *MnSPL16B*) were identified as miR156-targeted genes in mulberry in the present work. This finding implied that the *SPL* genes were the exclusive miR156 target genes across herbs and woody plants. The miR156/SPLs modules can also control the development of lateral roots and miR156-targeted *SPL10*-controlled root meristem activity in *A. thaliana* [29,33]. We also found three *MnSPL* genes (*MnSPL1*, *MnSPL13*, and *MnSPL14*) prominently expressed in the root, which implied that these genes might be involved in root development regulation.

As a conserved miRNA, miR172 participated in the regulation of flowering and was regulated by SBP-box transcription factors in *A. thaliana* [34,35]. Additionally, the miR172 expression level increased with age in *A. thaliana* and *P. × canadensis* [11,36]. In *A. thaliana*, *SPL9/10* activates the vegetative phase transition by promoting miR172 expression [3], while the over-expression of miR172 in *A. thaliana* can increase the expression of *SPL3*/*4*/*5* [37]. In mulberry, there were six mature miR172 transcripts, named mno-miR172a–f, and six *MnSPL* genes (*MnSPL2*, *MnSPL8*, *MnSPL10A*, *MnSPL10B*, *MnSPL15*, and *MnSPL16A*) were verified as the direct upstream regulators of mno-miR172a. Additional miR172-upstream *MnSPL* genes were identified in mulberry, which suggested there were more complex regulatory relationships between SPLs and miR172 in mulberry.

In summary, we found that the expression of miR156 subsequently reduced as mulberry aged, while *SPLs* and miR172 showed the opposite profile. A total of nine *MnSPL* genes as exclusive miR156 target genes were verified and six *MnSPL* genes could promote the expression of mno-miR172a in mulberry. The over-expression of miR156 reduced the expression levels of miR172 and *SPL* genes, except for two miR156 non-target genes (*MnSPL1* and *MnSPL12*) and decreasing the expression level of miR156 by STTM increased the miR172 and *SPL* genes expression levels, except for three non-target genes of miR156 (*MnSPL1*, *MnSPL12*, and *MnSPL14*). All of these experimental results indicated that the miR156/SPLs/miR172 regulation network also existed in mulberry, which implied this regulation pathway was conserved across angiosperms (Figure 8).

Our work should facilitate the integration of information about vegetative phase transition across perennial woody plants and give significant insight into how to reduce the breeding cycle of perennial woody plants.

## 4. Materials and Methods

### 4.1. Plant Materials and Growth Conditions

Tobacco seeds (*N. tabacum L*) were planted in sterilized soil and left at 4 °C for 2 d before transfer to a climate chamber at 25 °C under a long-day condition (16 h light/8 h dark), as were mulberry seedlings (*Morus atropurpurea* cv. Guiyou 62, abbreviated as GY62, which can bloom after 13 months). The older mulberry trees were grown on the open balcony of our laboratory. Plant age was measured from the date that seeds were transferred to the growth chamber.

Various mulberry tissues were used to analyze the expression of miRNAs and genes in this work. Young leaves, barks, and roots from 3-month-old (3 M), 11-month-old (11 M) and 16-month-old (16 M) mulberry were used to detect the expression of miR156 in mulberry by RT-qPCR. Young leaves from juvenile phase mulberry (J-YL, 3 months old) and young leaves of mature phase mulberry (M-YL, 16 months old) were used to analyze miR156 expression level by small RNA Northern blot assay. Young leaves of 2-month-old (2 M), 4-month-old (4 M), and 9-month-old (9 M) mulberry were used to detect the expression profile of miR172 in mulberry by RT-qPCR. Mature leaves from juvenile phase mulberry (J-ML) and mature leaves from mature phase mulberry (M-ML) were used to investigate the expression level of *MnSPLs*.

### 4.2. Bioinformatics Analysis of miR156, SPL Genes, and miR172 in Mulberry

We obtained sequence information from miR156s and miR172s in *Morus notabilis* from the NCBI/SRA database under accession number SRP032829. BLAST and HUMMER searches of *SPL* genes in MorusDB (https://morus.swu.edu.cn/hmmer, accessed on 3 March 2019) were carried out using *AtSPL* as the query sequence. An *SPL* gene structure diagram of mulberry was analyzed using Pfam (http://pfam.sanger.ac.uk, accessed on 19 March 2019) and GSDS 2.0 (http://gsds.cbi.pku.edu.cn/index.php, accessed on 19 March 2019). Additionally, 28 full-length *PtSPLs* [38], known *AtSPL* genes (TAIR, https://www.arabidopsis.org/, accessed on 30 March 2019), and 15 predicted *MnSPLs* were used to build a neighbor-joining (NJ) phylogenetic tree using MEGA5.1, and the branching reliability was assessed by the bootstrap re-sampling method using 1000 bootstrap replicates. The target genes of miR156 in mulberry were predicted by psRNATarget (http://plantgrn.noble.org/psRNATarget/, accessed on 3 March 2019) and the expectation value was limited to 4.0. GO analysis of predicted miR156 target genes was performed using the online software WEGO (https://wego.genomics.cn/, accessed on 23 December 2020).

### 4.3. Epression Analysis of miRNAs and Genes

Small RNAs (<200 bp) and total RNA were both extracted using the miRcute Plant miRNA Isolation Kit purchased from TIANGEN (DP504, Beijing, China, OD_260/280_ of all RNA samples were determined using a NanoDrop2000 (Thermo Scientific, Waltham, MA, USA), and agarose gel electrophoresis (AGE) was used to verify the integrity of RNA samples (data not shown). RNA probes were labeled with DIG by Riboprobe^®^ Systems (P1440, Promoga, Madison, WI, USA) and purified using an EasyPure^®^ RNA Purification Kit (ER701-1, TransGen Biotech, Beijing, China). Northern blots were performed as follows: twenty micrograms of small RNA were isolated on a 15% Ura-PAGE gel, transferred to a membrane using a Semi-Dry Transfer apparatus, cross-linked using ultraviolet crosslinking, and visualized using the protocol of a DIG DNA Labeling and Detection Kit (11093657910, Roche, Switzerland). Small RNA reverse transcriptions were performed using the miRNA First Strand cDNA Synthesis Kit (tailing reaction) (B532451, Sangon Biotech, Shanghai, China). The levels of mature miR156 and miR172 were quantified using the MicroRNAs qPCR Kit (SYBR Green method) (B532461, Sangon Biotech, Shanghai, China). *mno-miR166b* and *MnU6* were selected as reference genes together in our small RNA RT-qPCRs. Heatmaps were made using TBbtools [39], and the mimax normalization method was utilized to normalize this data. The PrimeScript™ RT reagent Kit with gDNA Eraser (perfect real time) (RR047Q, Takara, Japan) was used to reverse transcribe total RNA to cDNA. Quantitative reverse transcription polymerase chain reaction (RT-qPCR) was performed to analyze the transcript abundance of various *SPL* genes using the TB Green^®^ Premix Ex Taq™ II (Tli RNaseH Plus) (RR820Q, Takara, Japan). All primers and probes used in this work are listed in Appendix A.

### 4.4. Transient Trnsgene Expression in Mulberry

An 800-bp-predicted pre-miR156 sequence was cloned from the genome of GY62 into the pCAMBIA1301 vector to over-express mature miR156. Short tandem target mimicking (STTM) was performed to repress the expression of mature miR156. The STTM vectors were constructed as described previously [40,41]. For STTMd vector, three extra bases (cta) were inserted into the reverse complement sequences of mno-miR156c and mno-miR156d between the 11th and 12th bases. These two modified sequences were inserted into the end of 88-bp spacer sequence which could form the hairpin structure. pCAMBIA1301 was used to over-express this construct. The protocol of transient transgene expression of pCAMBIA-156 and pCAMBIA-STTM156cd in mulberry was as described in [42]. For GUS staining aay, samples were incubated with staining buffer (G3061, Solarbio, Beijing, China) at 37 °C for 12 h. The stained samples were washed by 75% (*v*/*v*) ethanol and photographed using a Leica microscope. All the primers and probes used in this work are listed in Appendix A.

### 4.5. Dual-Luciferase Repoter Assay

A 200-bp sequence containing the predicted cleavage of target *SPL* genes was cloned into the pGreenII 0800-LUC vector, which had the multiple cloning site after the 3′ end of the luciferase gene. The resultant vector was verified by sequencing. pCAMBIA-156 was used in dual-luciferase assays to over-express mature mno-miR156c. Vector structures are showed in (Figure 4c). Agrobacterium-mediated co-transformation of the pGreenII 0800-LUC and pCAMBIA-156 vectors into tobacco leaves was performed as described previously [43]. After infiltration fothree days, the ratio of LUC to REN activity was measured using the Dual-Luciferase Reporter Gene Assay Kit (11402ES80, Yeasen, Shanghai, China) on a configurable multi-mode microplate reader (Synergy™ H1, BioTek, Beijing, China). *MnSPL* geneloned into theGreenII 62-SK vector. A 1500-bp promoter sequence from mno-miR172a predicted by Promoter 2.0 Prediction Server (http://www.cbs.dtu.dk/services/Promoter/, accessed on 31 May 2020) was amplified and cloned into the pGreenII 0800-LUC vector. Vector structures are shown in (Figure 7c). Agrobacterium-mediated co-transformation and LUC/REN activity detection were processed as described above. All the primers and probes used in this work are listed in Appendix A.

### 4.6. Yeast One-Hybrid Assay

The fulllength sequences of *SPL* genes were amplified and then cloned into the pGADT7 vector. Because the full length of three genes were over 3000 bp, only partial sequences (1000 bp) from *Morus024784*, *Morus013868* and *Morus025152* were amplified. A 50-bp sequence containing a predicted *cis*-acting element was repeated three times and cloned into the pAbAi vector. A mutant *cis*-acting element (modified from GTACTGTAC to GATCTGATC) was used as a negative control. A verified concentration (150 ng/mL) of Aureobasidin A (AbA) was used to screen positive yeast strains in a yeast one-hybrid system. The assays were performed using the Matchmaker^®^ Gold Yeast One-Hybrid System (630491, Takara, Japan) according to the manufacturer’s instructions. All the primers and probes used in this work are listed in Appendix A.

### 4.7. Degradome Sequencing

We selectedroots, leaves, and bark from juvenile mulberry and winter buds from mature mulberry to constructed a degradome library. The purified cDNA library was sequenced on an Illumina HiSeq2000 Sequencing System (LC-BIO Sciences, Beijing, China).

## Figures and Tables

**Figure 1 ijms-22-05550-f001:**
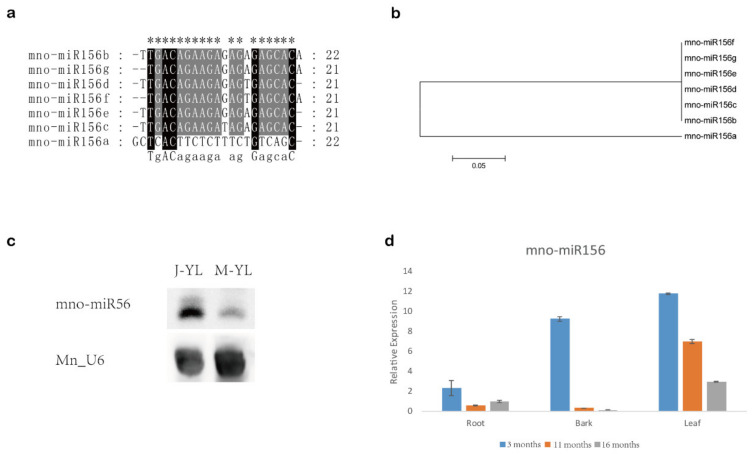
miR156 in mulberry. (**a**) Sequence alignment of miR156 in mulberry. “*” shows conserved sites. (**b**) Neighbor-joining (N.J.) phylogenetic tree of miR156 in mulberry. One thousand bootstrap replicates were used. (**c**) Northern blot showing the expression of miR156 in young leaves of juvenile (J.-Y.L.) and mature (M.-Y.L.) mulberry. Mn_U6 was used as a reference gene. (**d**) The expression profile of miR156 in three tissues (roots, bark, and leaves) at different time periods (3, 11, and 16 months) in mulberry. Values represent the mean ± SD from three biological replicates.

**Figure 2 ijms-22-05550-f002:**
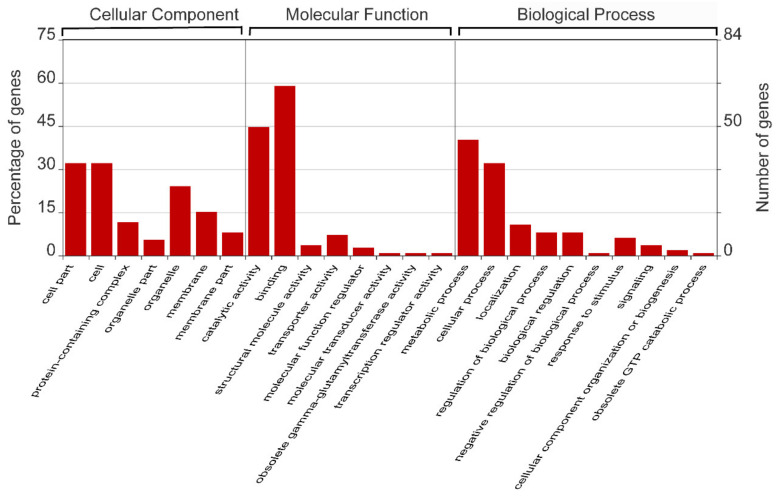
Gene ontology analysis of the predicted target genes of miR156 in mulberry.

**Figure 3 ijms-22-05550-f003:**
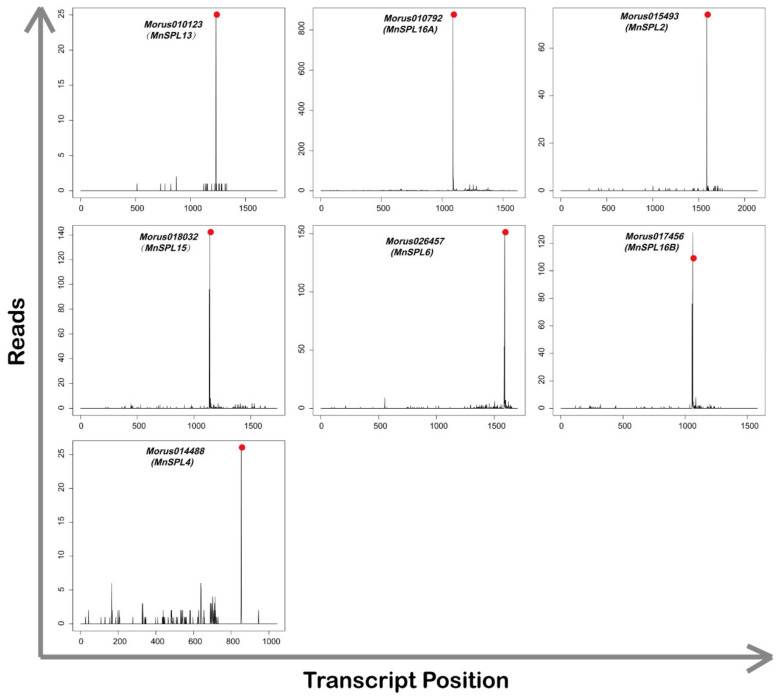
T-plot diagram of miR156 target genes verified by degradome sequencing. Red dots show the splice site of miR156 target genes.

**Figure 4 ijms-22-05550-f004:**
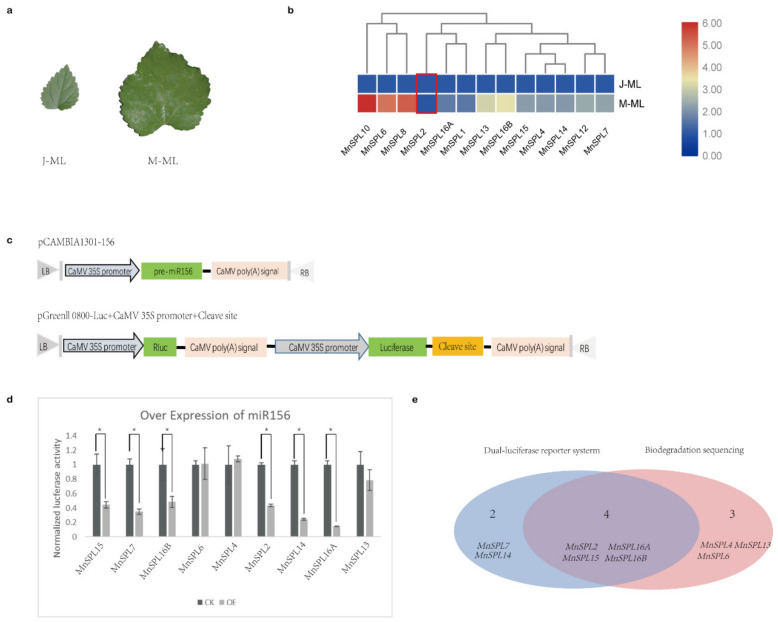
Identification of miR156 target genes in mulberry. (**a**) Morphological photos of mature leaves of juvenile (J-ML) and mature (M-ML) mulberry. (**b**) The expression trend of *MnSPLs* in mature leaves of juvenile (J-ML) and mature (M-ML) mulberry measured by RT-qPCR. Red boxes indicate there were no significant differences between this sample. Heatmap was drawn using TBtools, and the min–max normalization method was used to normalize the data. (**c**) Schematic of plasmids used in dual-luciferase assays. (**d**) The target genes of miR156 identified by dual-luciferase assays. CK: control check, empty pGreenII 0800-Luc vector and pCAMBIA-156; OE: reconstructed pGreenII 0800-Luc vector and pCAMBIA-156. Values represent the mean ± SD from three biological replicates and were statistically analyzed (independent-samples *t*-test): *, *p* < 0.05. (**e**) Venn diagrams showing miR156 target genes in mulberry.

**Figure 5 ijms-22-05550-f005:**
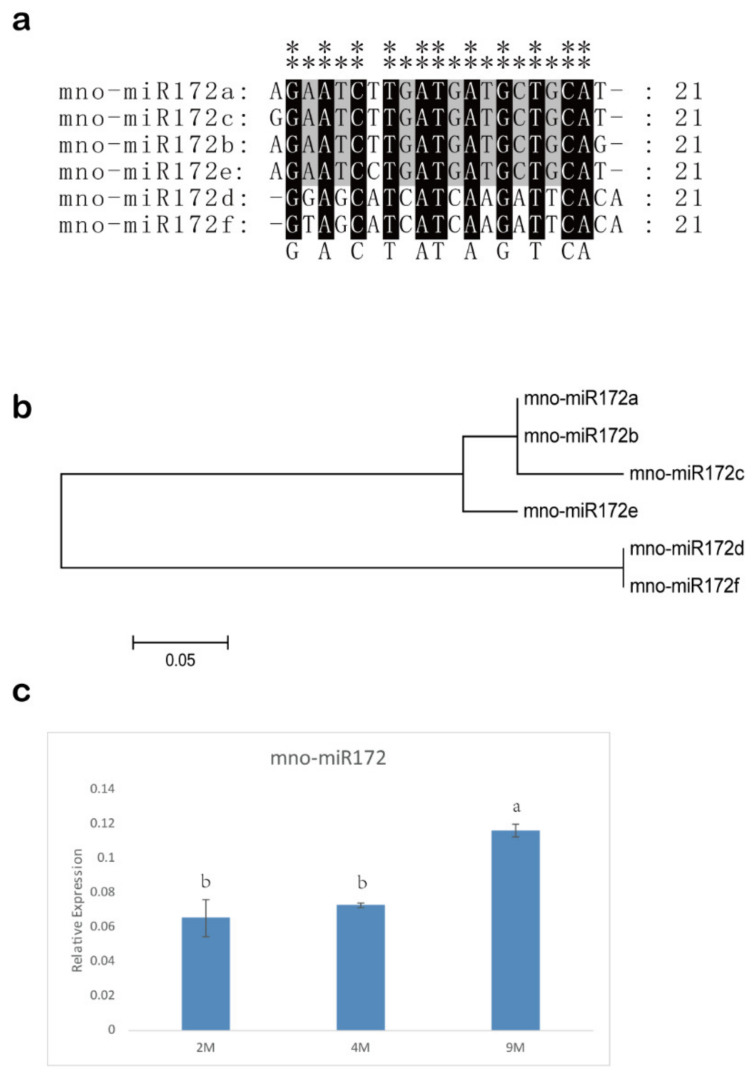
miR172 in mulberry. (**a**) Sequence alignment of miR1172 in mulberry. “**” shows highly conserved sites with no base difference, “*” indicates low conservation sites with two base differences. (**b**) Neighbor-joining (NJ) phylogenetic tree of miR172 in mulberry. One thousand bootstrap replicates were used. (**c**) The expression profile of miR172 in leaves at 2 months (2 M), 4 months (4 M) and 9 months (9 M) in mulberry. Values represent the mean ± SD of three biological replicates and were statistically analyzed (one-way ANOVA test): *p* < 0.05.

**Figure 6 ijms-22-05550-f006:**
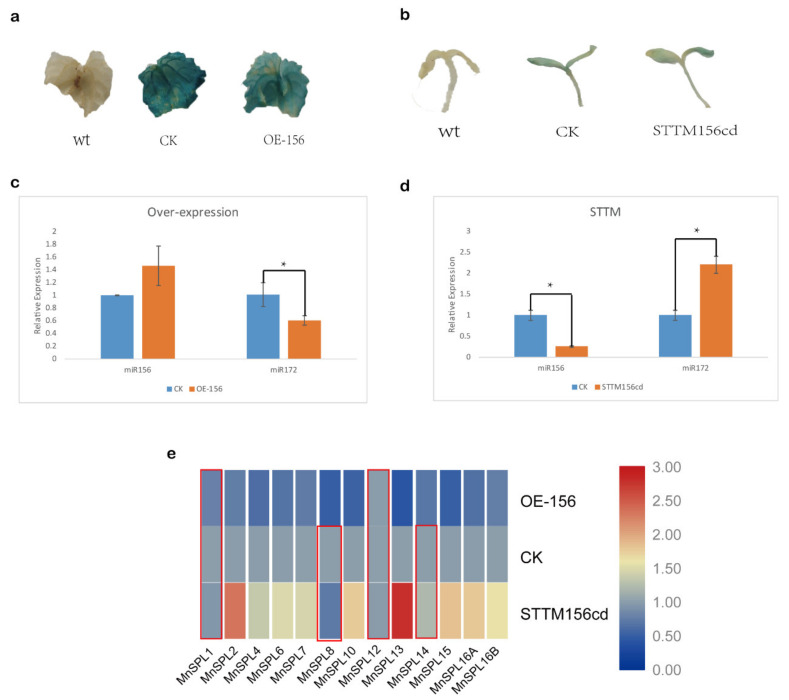
Altered miR156 expression in mulberry. (**a**,**b**) GUS staining of mulberry leaves and seedlings, wt: wild type; CK: control check, the empty pCAMBIA1301 vector; OE-156: over-expression of miR156; STTM 156cd: repression of mno-miR156c and mno-miR156d by short tandem target mimic (STTM). (**c**,**d**) The expression level of miR156 and miR172 under over-expression and reduced expression of miR156 in mulberry. Values represent the mean ± SD of three biological replicates and were statistically analyzed (independent-samples *t*-test): *, *p* < 0.05. (**e**) The expression profile of *SPL* genes under over-expression and reduced expression of miR156 in mulberry. Red box indicates there were no significant differences between these samples. Heatmap was finished using TBtools, and the min–max normalization method was utilized to normalize the data.

**Figure 7 ijms-22-05550-f007:**
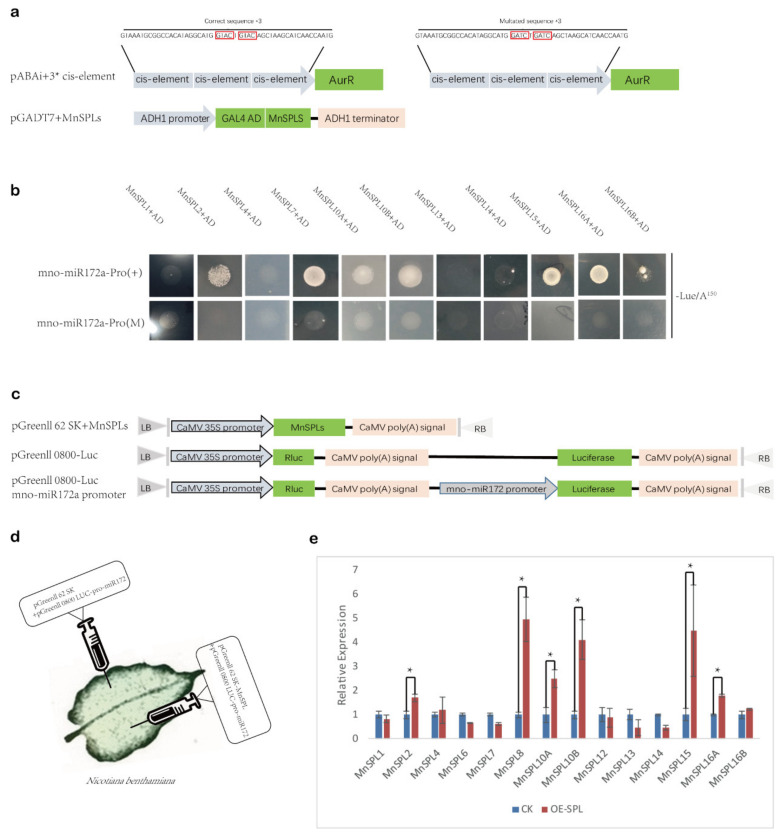
MnSPLs promote miR172 expression in mulberry. (**a**) Schematic of plasmids used in yeast one-hybrid assays. Red boxes show the sites of *cis*-elements in the promoter of miR172, which can be recognized by MnSPLs. (**b**) Yeast one-hybrid assay identified the interaction between MnSPLs and the promoter of miR172. (+) means a correct sequence of the *cis*-element. (M) means a mutated sequence of the *cis*-element. (**c**) Schematic of plasmids used in dual-luciferase assays. (**d**) Schematic diagram of dual-luciferase assays. (**e**) Dual-luciferase assays identified the interaction between MnSPLs and the promoter of miR172. CK: control check. OE-SPL: over-expression of *SPL* genes. Values represent the mean ± SD from three biological replicates and were statistically analyzed (independent-samples *t*-test): *, *p* < 0.05.

**Figure 8 ijms-22-05550-f008:**
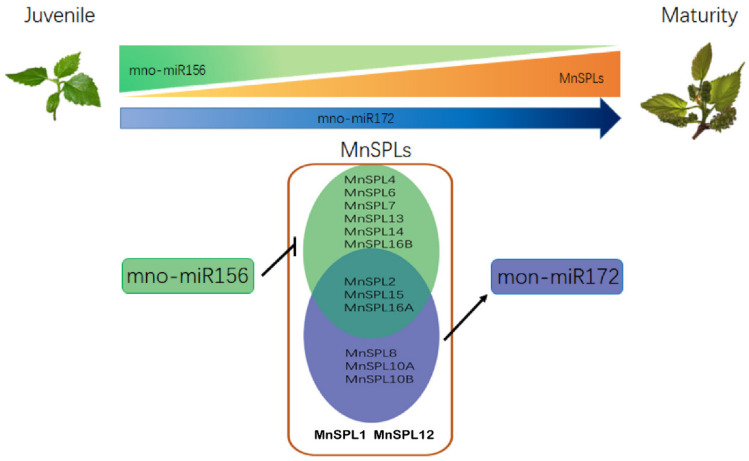
The molecular mechanism of miR156 in mulberry development. The larger area and darker color indicate higher gene expression level.

**Table 1 ijms-22-05550-t001:** miR156 and miR172 in mulberry.

Name	Sequence	Length	Read in Leaf	Read in Bark	Read in Male Flower
mno-miR156a	GCTCACTTCTCTTTCTGTCAGC	22	0	46	4
mno-miR156b	TTGACAGAAGAGAGAGAGCACA	22	6	2	337
mno-miR156c	TTGACAGAAGATAGAGAGCAC	21	12,787	117,270	768,488
mno-miR156d	TTGACAGAAGAGAGTGAGCAC	21	2942	14,574	8723
mno-miR156e	TTGACAGAAGAGAGAGAGCAC	21	2099	1277	155,199
mno-miR156f	TGACAGAAGAGAGTGAGCACA	21	22	378	65
mno-miR156g	TGACAGAAGAGAGAGAGCACA	21	8	4	1198
mno-miR172a	AGAATCTTGATGATGCTGCAT	21	19,265	8484	1245
mno-miR172b	AGAATCTTGATGATGCTGCAG	21	143	13	137
mno-miR172c	GGAATCTTGATGATGCTGCAT	21	15	266	6
mno-miR172d	GGAGCATCATCAAGATTCACA	21	11	6	4
mno-miR172e	AGAATCCTGATGATGCTGCAT	21	3	2	0
mno-miR172f	GTAGCATCATCAAGATTCACA	21	1	6	0

**Table 2 ijms-22-05550-t002:** Predicted target genes of miR156 in mulberry (Expectation ≤ 2.5).

Target Acc.	Expectation	Target Start	Target End	Alignment	Target Aligned Fragment	Inhibition	Molecular Function (Go)	Gene Family
Morus010792	0	1554	1574	:::::::::::::::::::::	GUGCUCUCUAUCUUCUGUCAA	Cleavage	Binding	SPL
Morus010123	1	2201	2221	::::::::::::::::::::	GUGCUCUCUAUCUUCUGUCAU	Cleavage	Binding	SPL
Morus015493	1.5	2135	2155	::::::::: :::::::::::	GUGCUCUCUCUCUUCUGUCAA	Cleavage	Binding	SPL
Morus018032	1.5	2589	2609	::::::::: :::::::::::	GUGCUCUCUCUCUUCUGUCAA	Cleavage	Binding	SPL
Morus014488	1.5	2018	2038	:::::::: :::::::::::	UUGCUCUCUCUCUUCUGUCAA	Cleavage	Binding	SPL
Morus021290	2	1586	1606	: .:::::::::::::::.:	UUUUUCUCUAUCUUCUGUCGA	Cleavage	/	/
Morus017456	2.5	1220	1240	::::::::: ::::::::::	GUGCUCUCUCUCUUCUGUCAU	Cleavage	Binding	SPL
Morus026457	2.5	2801	2821	::::::::: ::::::::::	GUGCUCUCUCUCUUCUGUCAU	Cleavage	Binding	SPL
Morus026689	2.5	549	569	:::::.:.:::.::::::.:.	GUGCUUUUUAUUUUCUGUUAG	Cleavage	Binding	WD40 repeat-like
Morus002371	2.5	3369	3389	:::::.:::::.:::: ::::	GUGCUUUCUAUUUUCUAUCAA	Cleavage	Binding	WD40 repeat-like
Morus011698	2.5	3694	3714	: :.:.:::.::::::::..:	GAGUUUUCUGUCUUCUGUUGA	Cleavage	Binding	WD40 repeat-like
Morus017876	2.5	3883	3903	:.:.:::.:.::::::.::	UAGUUUUCUGUUUUCUGUUAA	Cleavage	catalytic activity	FBP
Morus006566	2.5	1116	1136	:.:: .::::::.::::::	CCGUUCAUUAUCUUUUGUCAA	Cleavage	/	/

SPL: Squamosa promoter-binding-like protein. FBP: fructose 1,6-diphosphatase.

**Table 3 ijms-22-05550-t003:** Target genes of miR156 verified by degradome sequencing.

Gene_ID1	Gene_ID2	Gene Name	T_Start	T_Stop	T_Slice	D_RawReads	D_Pval
Morus010123	XM_010088675.2	*MnSPL13*	1221	1241	1232	25	0.00
Morus010792	XM_010114085.2	*MnSPL16A*	1078	1098	1089	875	0.00
Morus015493	XM_024169654.1	*MnSPL2*	1452	1472	1463	68	0.00
Morus018032	XM_010104258.2	*MnSPL15*	1129	1149	1140	142	0.00
Morus026457	XM_024175201.1	*MnSPL6*	1580	1600	1591	151	0.00
Morus017456	XM_024173253.1	*MnSPL16B*	758	778	769	140	0.01
Morus014488	XM_010109861.2	*MnSPL4*	842	862	853	26	0.01
Morus018550	XM_024174390.1	*MnGR-RBP7*	587	607	598	6	0.36
Morus012399	XM_010089645.2	*MnrsmG*	171	191	182	1	0.87
Morus014475	XM_024170153.1	*MnRH13*	1613	1640	1631	1	0.97
Morus021827	XM_024177195.1	*MnCABP5*	446	464	455	1	0.99

Gene_ID1: The ID of genes in Morus DB (https://morus.swu.edu.cn/, accessed on 3 March 2019). Gene_ID2: the ID of genes in NCBI (https://www.ncbi.nlm.nih.gov/, accessed on 3 March 2019). T_Start: the start site of predicted recognition region. T_Stop: the stop site of predicted recognition region. D_RawReads: the raw read of genes in degradome sequencing. D_Pval: the statistical *p* value of genes in degradome sequencing. SPL: squamosa promoter-binding-like protein. GR-RBP7: glycine-rich RNA-binding protein 7. rsmG: ribosomal RNA small subunit methyltransferase G. RH13: DEAD-box ATP-dependent RNA helicase 13. CABP5: calcium ion-binding protein.

## Data Availability

Publicly available datasets were analyzed in this study. This data can be found here: [SRP032829 in NCBI/SRA database, https://morus.swu.edu.cn/, https://www.arabidopsis.org/, http://www.phytozome.net/poplar.php.

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
