# Peer review of "Hierarchical Action of Mulberry miR156 in the Vegetative Phase Transition"

_ijms, 2021, doi:10.3390/ijms22115550_

Round 1

Reviewer 1 Report

The authors have performed several experiments to characterize the interplay among miR156, SPL genes and miR172 in mulberry with the aim to correlate it with phase transition of the tree species. They used several methods: bioinformatics, degradome sequencing, expression analysis, STTM analysis, dual-luciferase assays and yeast one hybrid. The extensive work is appreciated. However the paper is not written well also in terms of scientific language, therefore it is difficult to read and to get a clear message. A main improvement would come from professional support in scientific writing. The main Tables (1,2 and 3) are missing from the manuscript which makes the reading also less easy. In general the involvement of miR156, SPL genes and miR172 in phase transition is just hypotetical and is related to the expression in juvenile and mature tissues, therefore I would use more hypotetical sentences in the manuscript. The core of the results is based on bioinformatic and molecular data that identify mulberry SPL genes and their connection with either miR156 or miR172, there is no functional evidence of their involvement in phase transition. The authors mention over-expression and STTM of miR156 in mulberry plants (I suppose stable transgenics) but probably the phenotype will be a matter of future analysis. Nevertheless the molecular evidence obtained by the authors support the hypothesis of a conserved pathtway for phase transition involving the mentioned players.

Some suggestions:

Abstract

line 30: Additionally, dual luciferase reporter assays were carried out to validate these miR156-target MnSPL genes. The sentence is redundant with the following one, re-write or remove the current sentence.

 line 31: Furthermore, yeast one-hybrid system and dual luciferase reporter
assays revealed the details of six MnSPLs and their downstream factors (mno-miR172). "revealed details" is not very meaningful, re-write reporting the findings.

Introduction

line 52: People also found.. not very appropriate, re-write.

line 60: ref. 11 is missing from the reference list.

lines 80-99 a summary of the results is reported, the section is more suitable in discussion (as what is reported at lines 339-347), I think in the introduction is not so appropriate, a shorter paragraph with the aim of the work would be more appropriate.

Materials and methods

lines 109-114. the authors collect leaves from mulberry plants of different age (months), they define young and mature leaves, a better description for young and mature is required (for instance position on the branch etc.). They harvest: young leaves of juvenile phase (J-YL, 3 month old)...of mature phase mulberry (M-ML, 16 month old), line 110-111, then again at line 111-112  young leaves of 3 month old (3M) ...and 16 month old (16 M) mulberry, is it the same sampling? they report sampling of young leaves of 3 month old and 16 old months plants twice. In the results section they use leaves of 3 different ages (Fig.5c), it is not clear in which experiment the other samples are used.

line 154 please write more about the methods for STTM

line 156 cleave site, cleavage (also in other points)

line 157: the constructs were then verified, how? by sequencing?

line 158: how was the dual luciferase assay performed? did they use transient expression in mulberry leaves? there is a cartoon in Fig.7 but it is in the results, please add more details

the section Vector construction and transformation needs particular improvement; sub sections for each experiments and more details of the protocols would help. For instance when transient or stable transformation is used could be reported.

Results

All the main Tables are missing in the text (Table1 etc.)

Fig. 1 legend (lines 184-189) is too close to the manuscript text (starting again at line 190), it is confusing; it is the same for all the figures. Panel d reports years, but it is meant months

line 193: "indicated" is probably better replaced by "suggested"

line 196 line 197 line 202: predication? predicated?

line 203 values were less than 2.5, and found six of these genes belonging to the SPL family, three of which 204 were WD40 repeat-like genes, one was a fructose 1,6-diphosphatase gene and two genes with no  annotation. This sentence is unclear, do they mean that they found 6 genes with SPL binding motifs

line 245-247 Six highly conserved mno-miR172s were identified in mulberry, and the expression of mno-miR172a was significantly higher than other mno-miR172s (Figure 5a, b and Table 1). In Fig. 5 a and b  mno-miR172a and s are not reported, just mno-miR172, Table 1 is missing from the manuscript

line 257-258: Overexpression of miR156 in mulberry leaves was found to be significantly decreased the expression of miR172 and MnSPL genes except for MnSPL1 and MnSPL12; English is not correct, re-write

Fig.6 legend CK control check, what is it, explain compared to wild type. No GUS staining is reported in the materials and methods, also not in the results section, only in Fig.6a, this needs to be integrated and more detailed.

In general, considered the several experiments, the results are reported in a very brief manner, a more elaborated description would help the clarity.

Discussion

 line 310 miR156  expressed at higher levels in juvenile mulberry tissues than in mature mulberry tissues, which 311 implied that this conserved miRNA was also involved in the vegetative phase transition in mulberry. The sentence should be re-phrased into a more hypotetical manner.

In general re-writing is necessary for all the sections.

Reviewer 2 Report

In the manuscript entitled “Hierarchical action of mulberry miR156 in the vegetative phase transition” authors try to describe a corelation between mno-miR156, mno-miR172 and SPL proteins during vegetative phase transition. The text should be thoroughly analysed in terms of language correctness

Questions:

  • Materials and methods should be described more clearly and precisely
  • It is not possible to verify the quality of miRNA fraction using an agarose gel electrophoresis
  • Fig.1 d – years?
  •  

Author Response

Dear Editor,

We are resubmitting the revised manuscript (manuscript ID: ijms-1166087) entitled " Hierarchical action of mulberry miR156 in the vegetative phase transition" by Li et al again for publication in International Journal of Molecular Sciences. In this revised version, we have addressed all of the reviewers’ comments. We have thoroughly checked the language following the professional advice. In order for you to know which parts have been changed, we highlighted all of these rewritten sentences and words in gray. Now we are listing the detailed point-to-point response as follows.

We deeply appreciate your consideration of our manuscript and the helpful suggestions offered by reviewer. We have studied their comments carefully and made corrections. We hope you now find that this revised manuscript is suitable for publication.

Thank you in advance for your help.

Sincerely,

Ningjia He

Professor

State Key Laboratory of Silkworm Genome Biology, Southwest University, Beibei, Chongqing 400715, China

Tel: +86-23-6825-0797

Fax: +86-23-6825-1128

Response to Reviewer 2 Comments

Point 1: Materials and methods should be described more clearly and precisely

Response 1: As you suggested, we have provided a much more detailed assay process on the Materials and Methods section at lines 98-104, lines 145-153, lines 157-169 and lines 178-179.

Point 2: It is not possible to verify the quality of miRNA fraction using an agarose gel electrophoresis Fig.1 d – years?

Response 2: We verified the expression profile of mno-miRNAs by small RNA Northern blotting and RT-qPCR in this study. There was no agarose gel electrophoresis result exhibited in this work as shown in Fig.1. The “years” have been corrected to “months” in the panel of Fig.1 d.

 The revised manuscript was provided in the attachment.

Round 2

Reviewer 1 Report

The authors have revised the paper, but the are several inaccuracies.

Abstract: 

line 29 shown to "be" directly regulated

Introduction

line 72-73 re-write English is not correct

Materials and methods

line 98-104 concerning leaf age the author should not mention in which figure the tissue was used but in which experiment

line 141: transient TRANSGENE EXPRESSION in mulberry, also at line 149

line 148: which is incorrect

line 150 samples were incubated...I think for GUS staining, it should be mentioned

line 159, 167, 169: were is incorrect

line 188: miR156 IS differentially etc.

line 207 prediction

Discussion

line 323: confused is not appropriate, use another term

line 369: conserved

Reviewer 2 Report

I accept the explanation provided by the authors

Author Response

We would like to thank you very much for your valuable comments and good suggestions that greatly helped to improve our manuscript. Thank you very much for your time and efforts.